# Inhibition of Growth of TSC2-Null Cells by a PI3K/mTOR Inhibitor but Not by a Selective MNK1/2 Inhibitor

**DOI:** 10.3390/biom10010028

**Published:** 2019-12-24

**Authors:** Jilly F. Evans, Ryan W. Rue, Alexander R. Mukhitov, Kseniya Obraztsova, Carly J. Smith, Vera P. Krymskaya

**Affiliations:** Pulmonary, Allergy and Critical Care Division, Department of Medicine, University of Pennsylvania, Philadelphia, PA 19104-6118, USA; ryanrue@pennmedicine.upenn.edu (R.W.R.); mukhitov@pennmedicine.upenn.edu (A.R.M.); kobra@pennmedicine.upenn.edu (K.O.); cjs423@drexel.edu (C.J.S.); krymskay@pennmedicine.upenn.edu (V.P.K.)

**Keywords:** lymphangioleiomyomatosis (LAM), TSC2-null, rapamycin, S6K/S6, 4E-BP1/eIF4E, MNK1/2 inhibitor, PI3k/mTOR inhibitor

## Abstract

Lymphangioleiomyomatosis (LAM) is a rare metastatic cystic lung disease due to a mutation in a TSC tumor suppressor, resulting in hyperactive mTOR growth pathways. Sirolimus (rapamycin), an allosteric mTORC1 inhibitor, is a therapeutic option for women with LAM but it only maintains lung volume during treatment and does not provide benefit for all LAM patients. The two major mTORC1 protein synthesis pathways are via S6K/S6 or 4E-BP/eIF4E activation. We aimed to investigate rapamycin in combination with compounds that target associated growth pathways, with the potential to be additive to rapamycin. In this study we demonstrated that rapamycin, at a clinically tolerable concentration (10 nM), inhibited the phosphorylation of S6, but not the critical eIF4E releasing Thr 37/46 phosphorylation sites of 4E-BP1 in TSC2-deficient LAM-derived cells. We also characterized the abundant protein expression of peIF4E within LAM lesions. A selective MNK1/2 inhibitor eFT508 inhibited the phosphorylation of eIF4E but did not reduce TSC2-null cell growth. In contrast, a PI3K/mTOR inhibitor omipalisib blocked the phosphorylation of Akt and both S6K/S6 and 4E-BP/eIF4E branches, and additively decreased the growth of TSC2-null cells with rapamycin. Omipalisib, or another inhibitor of both major mTORC1 growth pathways and pAkt, might provide therapeutic options for TSC2-deficient cancers including, but not limited to, LAM.

## 1. Introduction

Lymphangioleiomyomatosis (LAM) cells are over-proliferative smooth muscle-like cells with TSC suppressor mutations that result in excessive activity of mTOR-driven growth [1,2,3,4]. Growth here refers to an increase in cell size and proliferation. The origin of LAM cells is still being debated but it is known that they are delivered to the lung via the lymph system and growth is enhanced by estrogenic stimuli [5]. mTOR is a PI3K-related protein kinase that forms the nucleus of two multiprotein complexes, namely mTORC1 and mTORC2 [6]. The mTORC1 complex is a central regulator of cell metabolism, cell cycle progression, autophagy, protein synthesis, and proliferation in response to environmental conditions [6] (Figure 1A). The allosteric mTORC1 inhibitor Sirolimus (rapamycin) has been approved for use in LAM after an outstanding collaboration between researchers, clinicians, and LAM patients to develop a therapeutic option for this rare disease [7]. However, the response to therapy is incomplete and some patients are intolerant of, or less sensitive to, Sirolimus [7]. Our aims in the present study were to investigate clinically acceptable compounds, with some differences when compared to rapamycin, with different targets and mechanisms of action than rapamycin, to define compounds that might have an additive therapeutic efficacy with rapamycin. In some cells rapamycin effectively blocks mTORC1 phosphorylation of S6K/S6 but, at clinically tolerable concentrations (5–25 nM) [8], it does not effectively block mTORC1 phosphorylation of the critical eIF4E binding residues (Thr37/46) on 4E-BP1 [9]. eIF4E is significantly less abundant than any other proteins in the eIF4F translation initiation complex and is rate-limiting for capped mRNA translation of proteins that progress cell cycling, growth, and proliferation [10]. In some cells, MAPK-interacting kinases (MNK1/2), phosphorylate eIF4E and enhance tumor growth [11,12]. Therefore, we hypothesized that if peIF4E increases the translation of proteins involved in the growth of LAM TSC2-null cells, then selective MNK inhibition might be beneficial in LAM. MNK1/2 inhibitors have been developed with acceptable clinical safety profiles, including tomivosertib (eFT508) [13] (Figure 1B). We showed that eFT508 completely inhibited peIF4E in TSC2-deficient LAM patient-derived angiomyolipoma cells but had no effect on their growth. Many dual catalytic mTOR inhibitors do not have tolerable clinical profiles, [14] so we investigated a clinically acceptable dual PI3K/mTOR inhibitor omipalisib (GSK2126458) [15] that has completed a Phase 1 clinical trial in patients with advanced solid tumors [16] and another Phase 1 trial in patients with idiopathic pulmonary fibrosis [17]. Our study demonstrated that omipalisib dose-dependently inhibits pAkt and both mTORC1 protein synthetic pathways and, in an additive manner with rapamycin, inhibited the growth of TSC2-null cells. Future studies with these compounds in LAM-derived organoids might help to define therapies tailored specifically for individual patients.

## 2. Materials and Methods

### 2.1. Compounds and Reagents

Tomivosertib (eFT508) was obtained from Selleck (Houston, TX, USA, Sirolimus (rapamycin) was obtained from LCL (Woburn, MA, USA), Torin1 was obtained from Cayman Chemicals (Ann Arbor, MI, USA) and omipalisib (GSK2126458) was obtained from MedKoo Sciences (Morrisville, NC, USA). Stocks were made in DMSO (Sigma-Aldrich, St. Louis, MO, USA) 1–100 mM and stored in aliquots at −20 °C, prior to appropriate dilution for inhibition studies. All other reagents were of the highest grade possible.

### 2.2. In Vitro Cell Culture

Human LAM patient-derived angiomyolipoma (AML) 621-102 TSC2-null cells were derived from a sporadic LAM-associated renal AML carrying a biallelic inactivation of TSC2 [5]; these were provided by Dr. Elisabeth Henske, Brigham and Women’s Hospital and Harvard Medical School, Boston, MA. Lung tissue was obtained from LAM patients under the National Disease Research Interchange (NDRI, Philadelphia, PA, USA)-approved protocols and LAM lung cell lines were derived, as previously described in [4,18]. Mouse Tsc2-null TTJ cells were obtained as described in [19] and TTJ-L cells were derived from these by the addition of luciferase and green fluorescent protein, with a retroviral construct. Cell lines were grown in DMEM media (Corning, Manassas, VA, USA) supplemented with 100 U/mL penicillin, 100 μg/mL streptomycin (Gibco, Gaithersburg, MA, NY, USA), 10 mM glutamine, and 10% fetal bovine serum (FBS) (Atlanta Biologicals, Atlanta, GA, USA) and were maintained at 37 °C in a humidified atmosphere with 5% CO_2_. For immunoblot analyses, the cells were plated at 3 × 10^5^ cells/well in 6-well plates in 10% FBS medium for 24 h. The medium was removed and the cells were washed with Dulbecco’s phosphate-buffered saline (DPBS) and medium containing 0.1% BSA, in place of 10% FBS added for 2 h. Appropriate drug dilutions were added (<0.1% DMSO final) and the incubation continued for 16–18 h. The cells were processed for immunoblot, as described in Section 2.4.

### 2.3. Cell Growth Assays

Cells were seeded in 96-well plates at 3 × 10^3^ cells/well in 200 μL 2.5% FBS complete medium. Prior to incubation at 37 °C, the plated cells were left at room temperature (RT) for 1 h, to distribute evenly. The compounds were added 24 h later (2 μL/200 μL) when the final concentration of DMSO was less than 0.1%, and the incubation was continued for 48 h. The cells were fixed by an addition of 20 μL of 25% glutaraldehyde Electron Microscopy Sciences (Hatfield, PA, USA) Crystal Violet (C3886) and then stained with 50 μL of 0.05% crystal violet (C3886, Sigma-Aldrich, St. Louis, MO, USA), in 25% methanol. Plates were immersed two consecutive times in water and were air-dried. Cells were lysed by addition of 200 μL of methanol and were mixed prior to the reading absorbance at 590 nm. Absorbance 590 correlated with the cell number.

### 2.4. Cell Protein Extraction and Immunoblotting

Cell monolayers were washed once with DPBS, then lysed on ice for 15 min in 100–200 μL RIPA cell lysis buffer (Sigma) supplemented with protease and phosphatase inhibitors (Roche, Brighton, MA, USA). Protein was determined by the BCA (Thermo Scientific, Waltham, MA, USA) assay and equal amounts of proteins were resolved by SDS denaturing polyacrylamide electrophoresis on 3–8% Tris-Acetate (large proteins), 4–12% Bis-Tris or 10–20% Tris-Glycine Novex gels (small proteins) (Invitrogen, Waltham, MA, USA), transferred to nitrocellulose via iBLOT, blocked with Tris HCL buffered saline (TBS) pH 7.4 LiCor Blocker, and incubated overnight at 4 °C, with primary antibody diluted in Blocker, 0.1% Tween 20. The blots were washed with TBS, 0.1% Tween 20, incubated for 1 h at RT with secondary antibody LiCor that was 800W-labeled and diluted at a ratio of 1:15,000 in Blocker, 0.2% Tween 20. It was then re-washed and dried and fluorescent image was acquired using an Odyssey IR imaging system (LiCor Biosciences Lincoln, NE, USA). Antibodies used for immunoblots, β-Actin, pS6 (Ser235/236), S6, TSC2, mTOR, pmTOR (Ser2448), p4E-BP1 (Thr37/46), 4EBP1, Akt, pAkt (Ser273), and β-Actin were all from Cell Signaling. Antibody to peIF4E (Ser209) (76256) was from Abcam (Woburn, MA, USA). All primary antibodies were used at 1:1,000 dilutions except β-Actin, which was at 1:15,000 dilution.

### 2.5. Lung and Cell Immunofluorescent Analyses

Immunofluorescent staining was performed on LAM patient lung tissue [18] or 621-102 TSC2-null cells. Lung tissue slides (10 μm) were deparaffinized using standard protocols for immunohistochemistry. 621-102 Tsc2-null cells were grown on glass coverslips and treated as for the 6-well immunoblot procedure, then fixed with 4% PFA, washed with PBS, and blocked with 1% BSA, 30 min before immunostaining. For antigen retrieval (peIF4E and IgG), slides were boiled 20 min in citrate buffer with 0.05% Tween 20 (pH 6.0) and then blocked for 20 min with 1% BSA. All primary antibodies were used at a 1:250 dilution. The cells were incubated in primary antibodies, namely pS6 (Ser235/236) or α-SMA (Cell Signaling, Danvers, MA, USA) or peIF4E (Ser209) Abcam (76256), for 1 h at RT. Lung tissue slides were incubated in primary antibody, overnight at 4 °C. Secondary antibodies were goat anti-mouse Alexa-594 (Abcam, #150116), goat anti-rabbit Alexa-594 (Abcam, #150080), and goat anti-rabbit Alexa-488 (Abcam, #150077). All secondary antibodies were used at 1:250 dilution. Cell or tissue slides were incubated with appropriate secondary antibody for 1 h at RT, then stained with DAPI (10 mM) (Sigma-Aldrich Gaithersburg, MA, USA) for 10 min, washed, and embedded in 80% glycerol. Samples were analyzed and pictures were taken with a Nikon Eclipse 2000 microscope (Nikon, Melville, NY, USA).

### 2.6. Statistical Analyses

Statistical analysis was performed using the GraphPad Prism 8.1 (GraphPad Software, San Diego, CA, USA). Paired experiments were analyzed by *t*-test with a *p*-value <0.05 being regarded as significant. Data are presented as mean +/− SD or +/− SEM.

## 3. Results

### 3.1. LAM Lung Cells Express peIF4E and Rapamycin Inhibits pS6 but Not p4E-BP1 or peIF4E

In multiple independent human LAM lung patient-derived cell lines, under the cell culture conditions used, rapamycin (10 nM) inhibited the phosphorylation of S6 (Ser235/236) but not the phosphorylation of 4E-BP1 (Thr37/46) (Figure 2A). Rapamycin was used at a concentration of 10 nM throughout our studies, since this is approximately the clinical blood concentration achieved in patients dosed once daily with 2 mg Sirolimus [8]. However, the pharmacokinetic/efficacy studies with Sirolimus were conducted in combination with other immunosuppressive drugs for transplantation patients with an effective dose range of 0.5–5 mg/day [20]. There has not been a comprehensive study of Sirolimus blood concentrations vs. efficacy in LAM patients. We used pS6 (Ser235/236) as the surrogate for activation of S6K protein synthetic pathways, since the immunoblot signal for pS6K in these LAM cell lines and in some of our TSC2-null cells was weak. In this 4–12% Bis-Tris Novex gel, all isoforms of 4E-BP1 ran as one band (Figure 2A), while in the following two immunoblot figures using 10–20% Tris-Glycine gradient gels, several isoforms of 4E-BP1 were resolved. p4E-BP1 (Thr37/46) and peIF4E (Ser209) were not reduced by rapamycin whereas pS6 (Ser235/236) was inhibited more than 90% (Figure 2B).

### 3.2. LAM Lung Cells Express peIF4E and the Selective MNK1/2 Inhibitor eFT508 Reduces peIF4E in LAM Angiomyolipoma Cells but Does Not Inhibit Cell Growth

Other researchers have shown the expression of pS6 and p4E-BP1 in LAM lung tissue [21]. We showed here, for the first time, that LAM lung lesions express peIF4E (Ser209) (Figure 2C). peIF4E was particularly abundant in spindle cells within LAM lesions (Figure 2D). peIF4E was co-expressed with pS6 and α-smooth muscle actin in LAM lesions (Figure 2E,F). As a control for non-specific immunofluorescence, we showed this area of the LAM lesion with the IgG control antibody that showed no specific staining (Figure 2G). In normal lungs, we also observed phosphorylated eIF4E but the immunofluorescent signal appeared to be less intense than in LAM lesions (data not shown).

In a TSC2-null LAM patient-derived angiomyolipoma cell line 621–102, pS6 and peIF4E were co- expressed mainly in the cytoplasm (Figure 3A) and the immunofluorescent signals could be inhibited by rapamycin (10 nM) plus eFT508 (1 μM) (Figure 3B). This treatment completely inhibited all pS6 immunofluorescence, while a small proportion of peIF4E appeared to remain in cytosol and nuclear bodies (Figure 3B). There was a negligible immunofluorescence background when the primary antibody was anti-IgG or when only Alexa 594 was added (Figure 3C). Treatment with rapamycin alone showed a complete inhibition of pS6 but no inhibition of peIF4E and the converse result was obtained with eFT508 alone (data not shown). Immunoblot analysis of the 621–102 cell lysates showed that after 16–18 h incubation with rapamycin (10 nM), pS6 (Ser235/236) was inhibited but not the total p4E-BP1(Thr37/46), although the relative expression of 4E-BP1 isoforms expressing pThr37/46 was altered (Figure 3D). The selective MNK1/2 inhibitor eFT508 (100 nM) blocked the phosphorylation of eIF4E and the combination of rapamycin plus eFT508, completely inhibited both pS6 and peIF4E (Figure 3D). In contrast, Torin1 (250 nM), a dual TOR kinase inhibitor, inhibited pS6 and p4E-BP1 but not peIF4E (Figure 3D). Expression of the ratios of immunoblot pS6/S6, p4E-BP1/4E-BP1, and peIF4E/eIF4E are shown in Figure 3E. Both Torin1 and rapamycin, but not eFT508, inhibited the growth of LAM patient-derived angiomyolipoma 621–102 cells (Figure 3F). The combination of rapamycin and eFT508 was non-significant from rapamycin alone, indicating the lack of reduction of growth by eFT508 at 500 nM, a concentration that completely inhibited phosphorylation of eIF4E. There was no significant additivity of eFT508 with rapamycin (Figure 3F). We observed similar immunoblot and growth inhibition results in mouse Tsc2-null TTJ-L cells (data not shown).

### 3.3. PI3K/mTOR Inhibitor Omipalisib Inhibits pS6, p4EB-P1, pAkt, and Cell Growth

In mouse Tsc2-null TTJ-L cells, we investigated the ability of a dual PI3K/mTOR inhibitor omipalisib to inhibit the phosphorylation of proteins subsequent to mTORC1 activation, including both S6K/S6 and 4E-BP1/eIF4E growth pathways (Figure 4A). All drug treatments resulted in 100% inhibition of pS6 (Ser235/236). Rapamycin (10 nM) did not inhibit total p4E-BP1 (Thr37/46) but increased the relative abundance of 4E-BP1 hypo-phosphorylated isoforms, yet still phosphorylated Thr37/46 at the critical eIF4E binding residues and increased pAkt (Ser473) and peIF4E (Ser209) (Figure 4A). Treatment with omipalisib (500 nM) resulted in 100% inhibition of pAkt and pS6 and ~50% inhibition of p4E-BP1. Omipalisib (5 μM) resulted in complete inhibition of pS6, p4E-BP1, and pAkt. We observed similar inhibition of phosphorylation by omipalisib in TSC2-null LAM patient-derived AML 621-102 cells (data not shown). Rapamycin (10 nM) or omipalisib (500 nM) alone reduced growth of TTJ-L cells but the compounds in combination additively inhibited cell growth (Figure 4B). By immunoblot analysis, omipalisib exhibited dose-dependent inhibition of pS6 (Ser 235/236), p4E-BP1 (Thr 37/46), and pAkt (Ser 473) (Figure 4C,D). Quantitation of immunoblot bands showed 50% and 100% inhibition of pS6 and pAkt at ~50 nM omipalisib, respectively. Inhibition of p4E-BP1 ~50% required 250–500 nM omipalisib (Figure 4C). In a correlative dose-dependent fashion, TTJ-L cell growth was inhibited 50% by ~250 nM omipalisib (Figure 4D).

## 4. Discussion

mTOR function is estimated to be hyperactive in up to 70% of all human tumors. LAM disease is a result of excessive proliferation through the mTOR growth pathways in TSC2-null LAM cells in the lung. Rapamycin is the only FDA-approved treatment for LAM disease but is not fully effective in all LAM patients and has a cytostatic rather than cytocidal effect on cells. Therefore, understanding the effects of mTORC1 pathway inhibition in TSC2-null cells subsequent to rapamycin treatment is important to tailor the therapy in the most effective manner. In addition to the increasing protein synthetic growth pathways, mTORC1 activation enhances lipid and nucleotide synthesis and decreases autophagy but many of these effects require translation of the effector proteins via the protein synthesis branches, following mTOR stimulation [6]. These are the ribosomal S6K/S6 pathway and the translation initiation 4E-BP1/eIF4E pathway [6]. Although well-known to the thought leaders in mTOR signaling, it is still underappreciated that rapamycin effectively inhibits the phosphorylation of the S6K/S6 growth arm but, at clinically achievable concentrations, does not inhibit the critical phosphorylation sites at Thr37/46 of 4E-BP1 in many but not all cells [9]. p4E-BP1 (Thr37/46) releases eIF4E, the rate-limiting mRNA translation initiation factor of the eIF4E complex, resulting in an active synthesis of selective proliferative and anti-apoptotic proteins. In LAM patient-derived lung cells, AML kidney-derived LAM cell lines and mouse Tsc2-null TTJ cells, we showed that rapamycin was an effective inhibitor of pS6 (Ser235/236) but a poor inhibitor of p4E-BP1 (Thr37/46). This is not true in all cell lines or in patients with other diseases. For example, it has been shown in T cells purified from lupus patients, treated with 2 mg daily rapamycin, about 50% p4E-BP1 (Thr37/46) was inhibited [22]. Indeed, rapamycin treatment has shown promise in several small clinical trials in autoimmune diseases, including systemic lupus erythematosus [23], rheumatoid arthritis, diffuse cutaneous scleroderma [24], and idiopathic multicentric Castleman disease [25]. Our in vitro cell studies did not investigate the effect of TSC2+/+ immune cells or stromal cells on the TSC2-null cell growth. We believe inhibition of growth of human LAM cell organoids, in the presence or absence of different human immune or stromal cells, would be more clinically relevant using concentrations of rapamycin that are clinically tolerable, i.e., between 5–25 nM [20].

MNK1/2 inhibitors of phosphorylation of eIF4E plus rapalogs have been shown to have an additive tumor growth inhibition in experimental models of glioblastoma, prostate, and lung cancer [26]. However, MNK1/2 activity does not appear to be essential for normal cell growth or development [27]. We showed here, for the first time, that there is a strong expression of peIF4E, co-localized with pS6 and α-SMA, in LAM lung lesions. We also demonstrated that the MNK1/2 inhibitor eFT508 completely blocked phosphorylation of eIF4E but did not inhibit growth of LAM patient-derived AML TSC2-null cells. In support of the concept that inhibition of phosphorylation of eIF4E is not linked to cell growth, under our conditions, the dual mTOR inhibitor Torin1 strongly reduced TSC2-null cell growth but had no effect on the phosphorylation of eIF4E. It is possible that in an in vivo TSC2−/− model, eFT508 might inhibit tumor growth, as was seen in an aggressive liver cancer model [26]. In the latter publication, eFT508 was shown to downregulate the translation of PDL-1 mRNA by the tumors in vivo and enhance survival profoundly [28]. Several studies have shown that either anti-PD-1 or anti-PD-L1 antibodies reduce TSC2-deficient tumor growth in vivo and enhance mouse survival [19,29]. Treatment with eFT508 in vivo might be additive with rapamycin or checkpoint inhibitors via the enhancement of the immune defense against TSC2-deficient tumor growth.

Many early phase cancer drugs have been developed as allosteric inhibitors of mTORC1 (rapamycin and rapalogs), ATP-competitive mTOR inhibitors, or dual PI3K/mTOR inhibitors [14]. Clinical tolerability is a key issue that needs to be addressed in these classes of inhibitors. Rapamycin has the drawback of preventing a negative feedback loop, via mTORC2 or S6K phosphorylation, resulting in increased pAkt [30]. Rapamycin increased phosphorylation of Akt (Ser473) in all our TSC2-deficient cells. We monitored Akt (Ser473) since after the initial activation of PI3K and the phosphorylation of Akt (Thr308), as a complete activation of Akt requires phosphorylation at Ser473 [30]. Torin1, an ATP-competitive mTOR inhibitor, prevented the phosphorylation of both S6 and 4E-BP1 but Torin is not tolerated in clinic. However, omipalisib a clinically acceptable PI3K/mTOR inhibitor, reduced phosphorylation of pS6 (Ser 235/236), p4E-BP1 (Thr37/46), and of pAkt (Ser473). Inhibition of pAkt, pS6, and p4E-BP1 by omipalisib, correlated with reduced cell growth. Importantly, the combination of omipalisib with rapamycin, additively inhibited TSC2-null cell growth. Whilst we have not investigated the different mechanisms of inhibition by omipalisib in our study, omipalisib has been shown to inhibit the phosphorylation of Akt (Thr308) and Ser (473), and to induce cell cycle arrest and inhibition of cell proliferation in many cancer cells [15].

In addition to estrogen [5], specific ligand stimulation of the insulin/Igf1/Igf2 and prolactin receptors, resulting in activation of multiple growth pathways including PI3K/Akt, might be a particularly important growth stimuli in LAM [31,32]. The cell of origin for LAM cells (TSC2−/−) and how a relatively small number of these cells can cause such widespread lung destruction is still not known. It is clear the effects of the compounds would also be seen on immune and stromal cells (TSC2+/+), where the TSC brake on mTORC1 would be possible. Therefore, mechanisms of action of inhibitors in the diseased lung might be very complex.

It is difficult and not very realistic in terms of time, to mimic multiple activation scenarios and titrate combinations of inhibitors in in vivo animal models. Since both rapamycin and omipalisib have preclinical and acceptable clinical profiles including tolerability, pharmacokinetics, and efficacy in different settings, and there are no primary LAM-cell derived tumors in animals, we aimed to develop primary LAM cell-derived organoids. These can be grown, in the presence and absence of specific stimuli and different stromal cell types, e.g., different types of immune cells, mesenchymal fibroblasts and epithelial cells, and the potency of inhibitors for growth inhibition can be determined for combination therapies like rapamycin and omipalisib. We believe, as others have suggested for various cancers, that patient-derived organoids might help define personalized therapy for LAM [33].

Our data emphasized the differential inhibition of pathways downstream of mTORC1 activation by rapamycin, as has been previously published in other cells [9]. Hayashi et al. [21] demonstrated in 30 LAM lung biopsies that there was a 93–97% high (2+/3+ IHC score) expression of pS6K/pS6 but only a 76% high expression of p4E-BP1. Neither eIF4E nor peIF4E were investigated in the latter study. However, in about 30% of cancers, elevated eIF4E correlated with a poor prognosis [34]. We hypothesized that some rapamycin-insensitive LAM patients might have an increased eIF4E/4E-BP1 ratio, making this mTORC1 activation arm more important in these patients [35].

In summary, while rapamycin is a potent inhibitor of the mTORC1-driven S6K/S6 pathway in LAM cells it is less effective against the tumorigenic 4E-BP/eIF4E pathway. Disappointingly, a selective MNK1/2 inhibitor blocked phosphorylation of eIF4E but did not inhibit TSC2-null cell growth. In contrast, a PI3K/mTOR inhibitor effectively reduced phosphorylation of S6, 4E-BP1, Akt and, in an additive manner with rapamycin, inhibited cell growth. We suggest that a clinically tolerable PI3K/mTOR inhibitor might be beneficial in LAM, as monotherapy or in combination with rapamycin, and in other mTORC1-driven neoplastic diseases. It has not escaped our attention that in age-related diseases, most studies on mTOR activation have emphasized the S6K/S6 arm [36] but we suggest that investigating the eIF4E/4E-BP ratio in aging tissues might uncover novel targets for the design of anti-aging therapeutics.

## Figures and Tables

**Figure 1 biomolecules-10-00028-f001:**
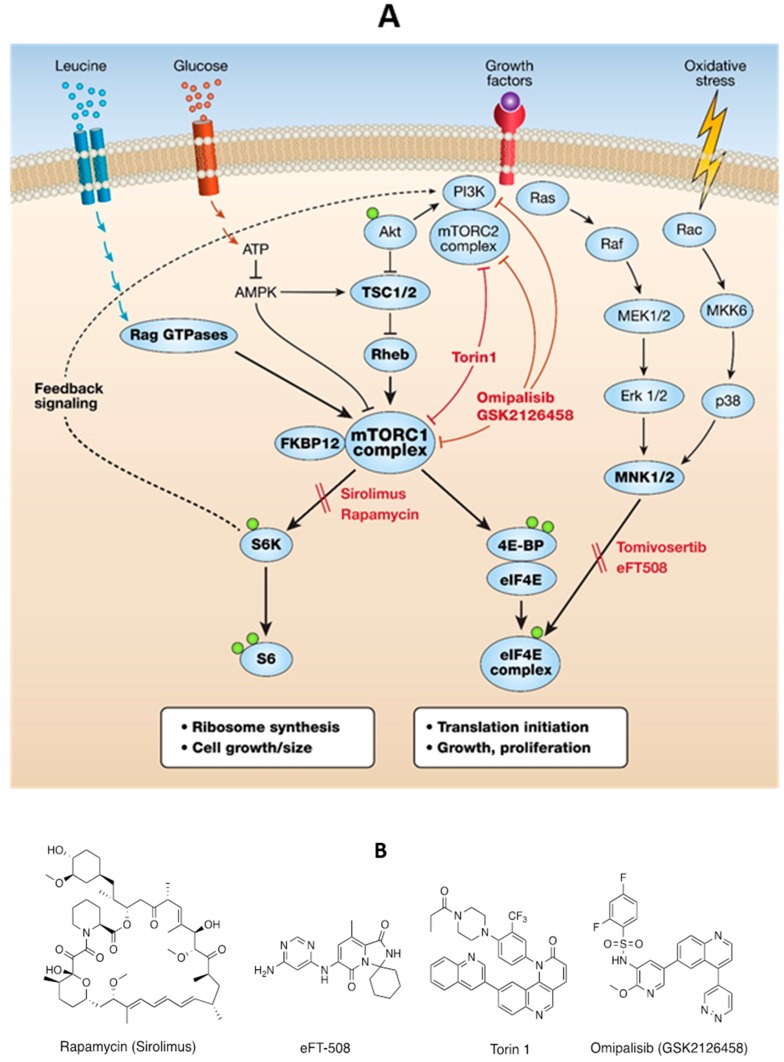
Schematic of mTORC1 growth pathways and inhibitors used in this study. (**A**) Cells respond to extracellular growth conditions by modulating mTOR activation. The mTORC1 complex is controlled by Rheb GTPase activation, which is regulated by TSC1/2 suppressor proteins. Oxidative stress or other external signals might activate MNK1/2 kinases, resulting in the phosphorylation of eIF4E. eIF4E is a critical protein in the eIF4F initiation complex for translation of cancer-related proteins. Green circles represent phosphorylation sites relevant to this study. Tomivosertib (eFT508) is a selective inhibitor of MNK1/2. Sirolimus (rapamycin) allosterically inhibits mTORC1, preferentially inhibiting the S6K/S6 pathway. Torin1 catalytically inhibits mTORC1 and reduces both the S6K/S6 and 4E-BP/eIF4E protein synthetic branches. Omipalisib (GSK2126458), a dual PI3K/mTOR inhibitor, inhibits the phosphorylation of Akt, mTOR, S6K/S6, and 4E-BP/eIF4E. (**B**). Chemical structures of compounds.

**Figure 2 biomolecules-10-00028-f002:**
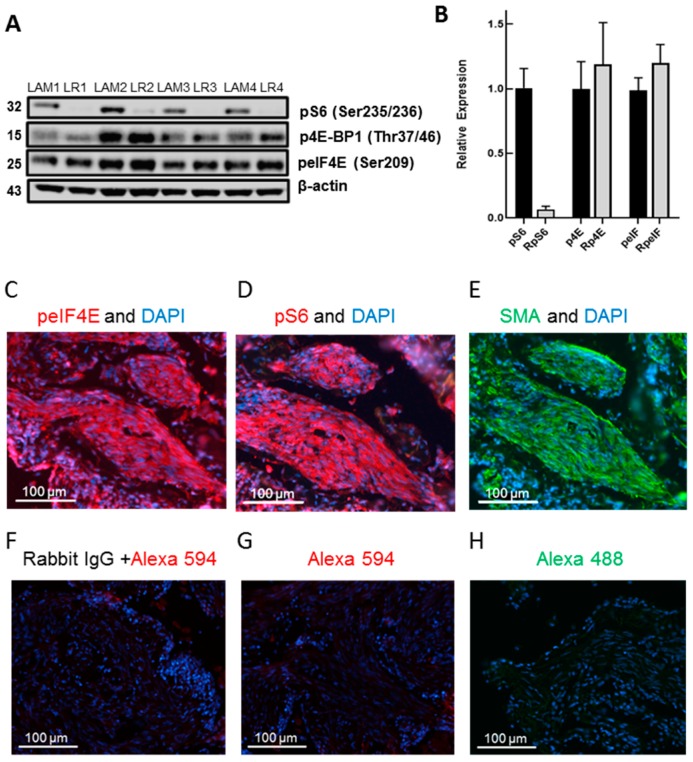
Rapamycin inhibition of pS6 but not p4E-BP1 or peIF4E in LAM lung-derived cell lines and peIF4E localization in LAM lung lesions. (**A**) Immunoblot of pS6 (Ser235/236), p4E-BP1 (Thr37/46), peIF4E (Ser209), and β-Actin in four independent LAM lung-derived cell lines (LAM1-LAM4), plus or minus rapamycin (10 nM) incubated for 16–18 h in a serum-deprived medium (LR1-LR4). Immunoblot analysis was as described in Materials and Methods. (**B**) Quantitation of the relative expression of immunoblot bands of pS6, p4E-BP1, and peIF4E by rapamycin-treated LAM lines relative to the control set for each protein control as one. LAM lung lesion immunofluorescent localization of (**C**) peIF4E (Ser209) (red), (**D**) pS6 (Ser235/236) (red), (**E**) α-SMA (green), (**F**) non-specific anti-IgG primary/Alexa 594 secondary (red), (**G**) Alexa 594 (red) secondary, and (**H**) Alexa 488 (green) secondary antibody.

**Figure 3 biomolecules-10-00028-f003:**
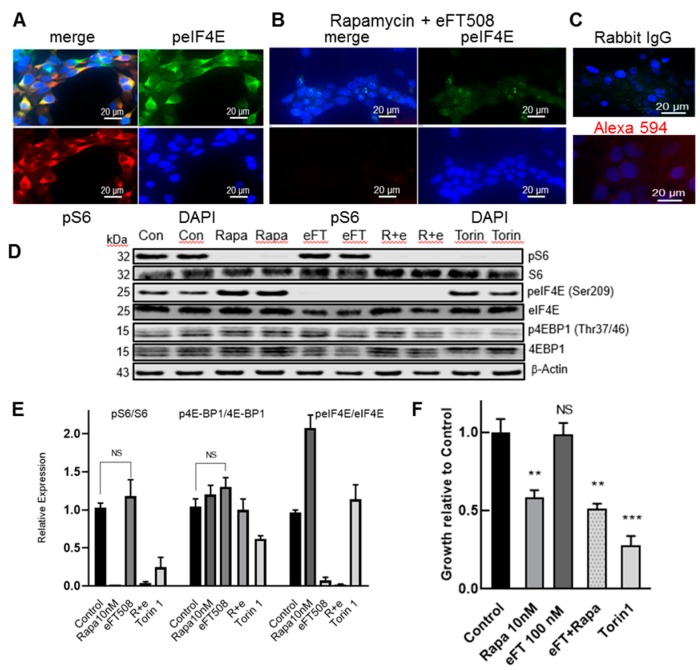
MNK1/2 inhibitor eFT508 inhibits peIF4E but not growth in TSC2-null LAM patient-derived angiomyolipoma cells. (**A**) Immunofluorescent analysis of LAM patient-derived AML 621-102 cells. (**A**) Merge (yellow) of pS6 (Ser235/236) (red) and peIF4E (Ser209) (green), peIF4E alone (green), pS6 alone (red), and 2-(4-amidinophenyl)-1H-indole-6-carboxamidine (DAPI) DNA-stained blue nuclei. All are rabbit primary antibodies. (**B**) Pretreatment of 621-102 cells with Rapamycin (10 nM), plus eFT508 (1 μM) peIF4E (green), pS6 (red), and DAPI (blue). (**C**) Control anti-rabbit IgG Alexa 488 and Alexa 594. (**D**) Immunoblot detection of pS6, S6, peIF4E, eIF4E, p4E-BP1, 4E-BP1, and β-Actin in 621-102 cells incubated for 16–18 h in a serum-deprived medium. The immunoblot represents data from three independent experiments with duplicate or triplicate samples. (**E**) Expression of pS6/S6, p4E-BP1/4E-BP1, and peIF4E/eIF4E corrected for β-Actin expression in each lane and relative to the appropriate control, set as one. Error was calculated +/− S.E.M and NS = Non-significant. (**F**) Growth inhibition of 621-102 cells, incubated in a medium containing 2.5% serum in 96 wells for 48 h, by rapamycin (10 nM) (Rapa) and Torin (500 nM), but not by eFT508 (100 nM) (eFT) and in combination (R + e). Cell growth was determined as described in Materials and Methods. Control growth absorbance 590 was set as one and drug inhibition was graphed relative to control. Graph F is representative of 4 independent studies in quadruplicate. Error was calculated +/− SD and significance relative to control is shown as ** *p* < 0.01; *** *p* < 0.001 or NS—non-significant.

**Figure 4 biomolecules-10-00028-f004:**
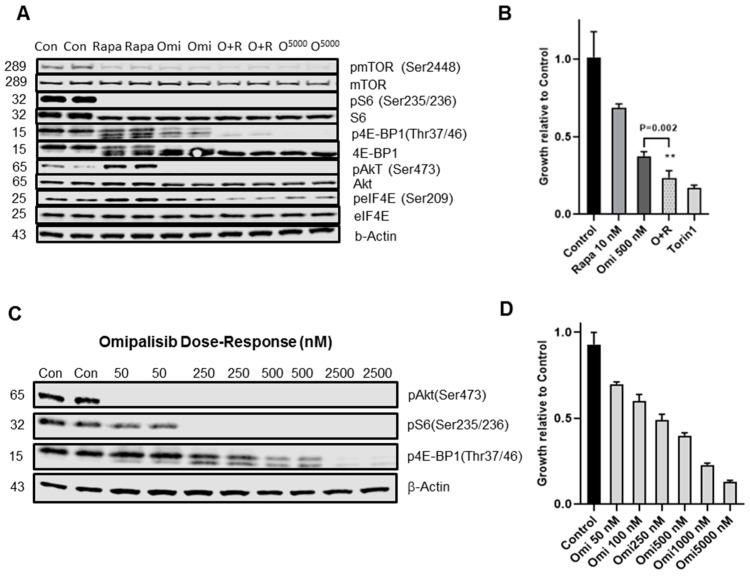
Omipalisib inhibition of pS6, p4E-BP1, and growth in TSC2-null TTJ-L cells. (**A**) TTJ-L cells were grown in the presence or absence of compounds for 16–18 h in a serum-deprived medium and the proteins were identified by immunoblot, as described in Materials and Methods. Rapa—Rapamycin (10 nM), Omi—Omipalisib (500 nM), O + R—omipalisib (500 nM) plus rapamycin (10 nM) and O2500—omipalisib (2500 nM). The blot represents 2 independent experiments in duplicates. (**B**) Inhibition of growth of TTJ-L cells, incubated in medium containing 2.5% serum in 96 wells for 48 h, by rapamycin (10 nM), omipalisib (500 nM), O + R omipalisib (500 nM) plus rapamycin (10 nM), or Torin1 (500 nM). Data expressed relative to the control set as one. Graph is representative of 2 independent growth studies. The difference of growth inhibition by Omi vs. O+R is given +/− SD ** *p* = 0.002. (**C**) Immunoblot of dose-dependent inhibition of pS6, p4E-BP1, and pAkt by omipalisib. (**D**) Dose-dependent inhibition of growth of TTJ-L cells by omipalisib.

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
