# Peer review of "Inhibition of Growth of TSC2-Null Cells by a PI3K/mTOR Inhibitor but Not by a Selective MNK1/2 Inhibitor"

_biomolecules, 2019, doi:10.3390/biom10010028_

Round 1

Reviewer 1 Report

The paper by Evans et al., describes the activity of the mTOR/MNK1/2 inhibitors on tuberos sclerosis 2 null cells, and suggests new directions in the therapeutic considerations for the management of LAM and aging related disorders. Considering that these molecules tested have been exploited therapeutically, this is a topic worth pursuing considering the plethora of data available in targeting components of the mTOR central signaling in disease and health. However, several major and minor aspects dampens the enthusiasm and conclusion of this work. Should the address these concerns, this will increase the readability and visibility of this work when published.

Major:

Authors should provide the translational relevance of their work either using In vivo model of the disease using the TSC2-null and control cells or at least a three-dimensional spheroid or model of the disease. The biological relevance on the biology of the cells should be analyzed including apoptosis and wound healing and to show the consequences of such treatment in the immediate upstream target of the specific inhibitors. Authors show present phase-contrast photomicrographs including close-ups of the control and treated cells in addition to the suboptimal immunofluorescent images provided. What are the cell cycle components targeted in common vs dissimilarly by the inhibitors. Authors should provide total protein expressions of 4EBP,S6, Elf4e etc.. in figure 2 as similarly shown in figure 3,4 etc..

Minor:

Authors should indicate what the expression levels of other tarets such as the AKT T308 are in similar blots, as does the S473 and discuss their relevance in relation to the therapeutic goal and the objective of the study.

Reviewer 2 Report

This study shows differential effects of omipalisib and rapamycin on the growth and the phosphorylation of mTOR substrates in TSC2-deficient cells in vitro. The experiments are generally well controlled and well presented. From the perspective of this reviewer the discussion can be improved since a failure to inhibit 4E-BP1 phosphorylation by rapamycin in the paper is not a general feature of its ability in blocking of mTORC1. For example, treatment of patients with lupus with rapamycin in vivo blocked the phosphorylation of both S6K and 4E-BP1 (J. Immunol. 182: 2063-2073, 2009).

This paper nicely shows that omipalisib inhibits both mTORC1 and mTORC2 which is not seen when treating cells with rapamycin (Figure 4). However, the figure legend or the results section fails to mention the duration of treatment with the drugs. This is critical since rapamycin only blocks mTORC1 in vitro after treatment for 24 hours, however, long-term treatment blocks mTORC2 (Mol Cell. 2006 Apr 21;22(2):159-68).

It would be important to state the duration of drug treatments and preferably compare them after exposures for 24 hours versus 2 weeks.

Such dual blockade of mTORC1 and mTORC2 by rapamycin appears relevant for the expansion of Tregs and therapeutic efficacy in lupus (Lancet, 391:1186-1196).

Therefore, the authors should discuss such clinical/medical relevance of their findings for lupus and other conditions of dual mTORC1/mTORC2 activation.

Author Response

Please see atachment

Round 2

Reviewer 1 Report

The authors have clearly articulated their responses to my concerns with appropriate caveats.

However, some minor concerns still persist in the current version.

page 4 line 144 please provide the data not shown as supplementary data 2.page 5. please provide better resolution images for giure 2C,D and E. Page 8. please provide enhanced, eligible and clear images of figure 4. 

Author Response

Second response to reviewer 1.

Reviewers comments:

page 4 line 144 please provide the data not shown as supplementary data 2. page 5.

Please provide better resolution images for Figure 2C, D and E. Page 8. Please provide enhanced, eligible and clear images of figure 4. 

Response to reviewer 1

We have been unable to confirm an increase in pAkt in the LAM lines specifically used in Figure 2A so we have removed the following sentence “ We also demonstrated that rapamycin increased the phosphorylation of Akt ~two-fold as has been previously been observed in other cells lines(data not shown) [20]. On Line 142 p4

We have provided better resolution of all Figures except Fig 1 A which already had the best resolution. We thank reviewer 1 for pointing this out. The images are much sharper now.

Reviewer 2 Report

The authors failed to address the medical and clinical relevance of their findings for lupus and other conditions of dual mTORC1/mTORC2 activation. Such discussion could enhance the impact of the in vitro studies conducted in this manuscript. Dual blockade of mTORC1 and mTORC2 by rapamycin appears relevant for the expansion of Tregs (Arthritis Rheumatol. 70:427-438; 2018) and therapeutic efficacy in patients lupus in vivo (Lancet, 391:1186-1196; 2018).

Author Response

Reviewer 2 Second Review: The authors failed to address the medical and clinical relevance of their findings for lupus and other conditions of dual mTORC1/mTORC2 activation. Such discussion could enhance the impact of the in vitro studies conducted in this manuscript. Dual blockade of mTORC1 and mTORC2 by rapamycin appears relevant for the expansion of Tregs (Arthritis Rheumatol. 70:427-438; 2018) and therapeutic efficacy in patients lupus in vivo (Lancet, 391:1186-1196; 2018).

Our response to Reviewer 2

We agree with this reviewer, and apologize to him or her, that we have not been able to do justice to the important roles of mTOR/rapamycin in autoimmune diseases. We have added an extra 3 autoimmune mTOR/rapamycin papers.

P9. Line 280-287Indeed, rapamycin treatment has shown promise in several small clinical trials in autoimmune diseases including systemic lupus erythematosus [23], rheumatoid arthritis, diffuse cutaneous scleroderma [24] and idiopathic multicentric Castleman disease [25]. Our in vitro cell studies do not investigate the effect of TSC2+/+ immune cells or stromal cells on the TSC2-null cell growth. We believe inhibition of growth of human LAM cell organoids, in the presence or absence of different human immune or stromal cells, would be more clinically relevant, but only using concentrations of rapamycin that are clinically tolerable i.e. between 5-15 nM.

The new reference 23 is the Lancet SLE reference kindly suggested by this reviewer. The new 24 is a very recent in press review including clinical trials on RA and SSc, and I have added a paper on the devastating autoimmune disease, Castleman disease, since I know the first author David Fajgenbaum whose life has been “saved” by rapamycin.

[23]   Lai, Z-W.; Kelly, R.; Winans, T, et al. Sirolimus in patients with clinically active systemic lupus erythematosus resistant to, or intolerant of, conventional medications: a single-arm, open-label, phase 1/2 trial. Lancet (2018) 391, 1186-1196.

[24]   Suto, T.; Karonitsch, T. The immunobiology of mTOR in autoimmunity. J Autoimmunity (2019) 102373

[25]   Fajgenbaum, D.C.; Langan, R-A.; Sada Japp, A., et al. Identifying and targeting pathogenic PI3K/AKT/mTOR signaling in IL-6 blockade-refractory idiopathic multicentric Castleman disease. J. Clin. Invest. (209)129, 4451-4463.

We have also added the need to investigate immune cell additions to LAM organoid to help translate to clinic. Line 330.   Lam Organoids grown………in the presence and absence of different sub-populations of immune cells…….